# Cationic Nanoparticle-Based Cancer Vaccines

**DOI:** 10.3390/pharmaceutics13050596

**Published:** 2021-04-21

**Authors:** Jeroen Heuts, Wim Jiskoot, Ferry Ossendorp, Koen van der Maaden

**Affiliations:** 1Department of Immunology, Leiden University Medical Centre, 2300 RC Leiden, The Netherlands; J.M.M.Heuts@lumc.nl (J.H.); F.A.Ossendorp@lumc.nl (F.O.); 2Division of BioTherapeutics, Leiden Academic Centre for Drug Research (LACDR), Leiden University, 2300 RA Leiden, The Netherlands; w.jiskoot@lacdr.leidenuniv.nl; 3TECO Development GmbH, 53359 Rheinbach, Germany

**Keywords:** cancer, immunotherapy, vaccine, liposomes, nanoparticles, polymers

## Abstract

Cationic nanoparticles have been shown to be surprisingly effective as cancer vaccine vehicles in preclinical and clinical studies. Cationic nanoparticles deliver tumor-associated antigens to dendritic cells and induce immune activation, resulting in strong antigen-specific cellular immune responses, as shown for a wide variety of vaccine candidates. In this review, we discuss the relation between the cationic nature of nanoparticles and the efficacy of cancer immunotherapy. Multiple types of lipid- and polymer-based cationic nanoparticulate cancer vaccines with various antigen types (e.g., mRNA, DNA, peptides and proteins) and adjuvants are described. Furthermore, we focus on the types of cationic nanoparticles used for T-cell induction, especially in the context of therapeutic cancer vaccination. We discuss different cationic nanoparticulate vaccines, molecular mechanisms of adjuvanticity and biodistribution profiles upon administration via different routes. Finally, we discuss the perspectives of cationic nanoparticulate vaccines for improving immunotherapy of cancer.

## 1. Introduction

Cancer immunotherapy, defined as the ability to mobilize the host’s immune system to kill cancer, has recently taken a central role within mainstream oncology and has shown unprecedented clinical responses in patients, coinciding with the development of novel classes of immunotherapeutic drugs [1,2,3,4]. Cancer-specific T-cells can be present in patients with various cancer types, but these T-cells are normally suppressed due to the immunosuppressive tumor microenvironment. The development and application of immune checkpoint inhibitors, which are antibody-based drugs that block suppressive immune signals for T-cells, revealed the enormous potential of these tumor-specific T-cells in the treatment of cancer [1,2]. Despite the success of these checkpoint inhibition therapies, still only a limited number of patients fully benefit from it (circa 20%), while treatment can result in severe side effects, such as autoimmunity [1,5,6,7]. This shows the need for immunotherapies that can induce high numbers of effective and functional tumor-specific T-cells, without inducing immune-related adverse events [8]. This can be done by therapeutic vaccination, which in contrast to prophylactic vaccination, aims to destroy cancer mainly via antigen-specific T-cells.

In the last decades, therapeutic cancer vaccines have proven to induce T-cells capable of achieving tumor regression without inducing severe immune-related adverse events, thereby offering highly specific cancer immunotherapy [9,10,11,12,13,14,15,16,17,18,19,20]. In addition, cancer vaccines are an efficient tool to amplify and diversify the repertoire of tumor-specific T-cells, which in turn could facilitate tumor regression. For cancer vaccines to work, tumor antigens need to be delivered to dendritic cells (DCs) which in turn can process and present antigen-derived peptides via MHC class I and II molecules to naïve CD8^+^ and CD4^+^ T-cells respectively, and activate these T-cells to proliferate [2,11,20,21]. The first step to induce an effective tumor-specific T-cell response is by adequately delivering the antigen to DCs and subsequently activating these antigen-presenting cells. This can either be achieved by direct vaccination in vivo or via stimulation of autologous DCs ex vivo and using these antigen-loaded cells as a cellular vaccine (the latter is reviewed elsewhere) [17,20,22,23,24]. For direct in vivo vaccination approaches, a large variety of delivery vehicles and adjuvants have been developed and investigated in combination with a multitude of tumor antigens. Despite these efforts, it has proven to be difficult to induce high numbers of functional tumor-specific T-cells in cancer patients [9,20,25]. During the past decades, a large number of nanoparticle types have been developed to target DCs and induce cellular immune responses. Among these nanoparticles, cationic particles are of special interest, because they have shown to have superior immunostimulatory properties as compared to their neutral and anionic analogues and have proven to be potent inducers of antigen-specific T-cells [9,26,27,28,29,30,31,32,33,34,35]. Recent preclinical and clinical studies have shown that cationic nanoparticles offer clinically applicable vaccine formulation platforms [9,10,36,37,38].

In this review, we discuss the application of cationic nanoparticles in cancer vaccine candidates and their role as formulation adjuvant. Besides, we discuss the adjuvant mechanism of cationic nanoparticles from the moment of injection, the biodistribution and uptake by antigen presenting cells (APCs) to the final induction of cancer-specific T-cells. First, the different types of cationic nanoparticles for cancer vaccination in combination with various types of tumor antigens are reviewed. Next, we will focus on the biodistribution profiles related to the route of administration. Finally, we discuss several molecular mechanisms via which cationic nanoparticles can enhance the efficacy of cancer vaccines.

## 2. Cationic Nanoparticles in Cancer Vaccine Design

### 2.1. Cationic Nanoparticles Improve Vaccine Efficacy

Cationic nanoparticles have been studied for a large variety of applications, such as prophylactic vaccines, therapeutic vaccines and for the transfection of cells and organisms with genetic material. For most vaccines cytosolic antigen delivery is sufficient, while for plasmid DNA the transfection requires more complex nuclear delivery [39,40]. A wide variety of cationic nanoparticle-based vaccines against viruses, bacteria and fungi to induce humoral (B-cell mediated antibody) responses have been studied in preclinical and clinical research. Additionally, antigen-associated cationic nanoparticles have also been investigated in a multitude of therapeutic vaccines directed against intracellular pathogens, which aim to induce a cellular immune response [10,26,37,38,41,42,43,44,45,46,47,48,49,50,51,52], and are discussed in detail below.

Cancer vaccines are aimed to elicit immune responses directed against tumor antigens that can either be non-tumor-specific (e.g., over-expressed self-antigens), tumor-associated (e.g., embryonal or tumor testis self-antigens) or tumor-specific (oncovirus induced or coded by specific DNA mutations or neo-antigens). In these cases, the tumor antigens can be intracellularly located, and therefore such cancer vaccines should induce a cellular immune response due to the nature of antigen recognition by T-cells. This class of immune cells is able to see intracellularly derived processed peptides, containing the antigenic epitopes, presented in MHC molecules at the cell surface. During the past decades, multiple strategies have been reported via which nanoparticles increase the immunogenicity of cancer vaccines: efficient uptake by DCs, immunostimulating properties (e.g., induction of cytokine production, upregulation co-stimulatory molecules) and depot formation at the side of injection upon vaccine administration (prolonging antigen exposure) [28,53,54,55,56,57,58,59,60]. Hereby, nanoparticles can mediate vaccine delivery through in vivo barriers (e.g., cell membranes and lysosomes) and prevent nucleic acid degradation.

Especially, cationic nanoparticulate formulations seem very promising, since they have shown to efficiently induce cellular immune responses [9,26,29,30,33,34]. Besides, direct comparison of three cationic nanoparticles (liposomes, chitosan coated PLGA- and maltodextrin-based particles) to their anionic equivalents showed superior intracellular protein delivery for the cationic nanoparticles [54]. Cationic nanoparticles have also shown to efficiently deliver mRNA-based vaccines in vivo [9,61,62,63,64,65]. Finally, in a direct comparison between peptide-loaded anionic and cationic nanoparticulate vaccines, the cationic nanoparticles induced stronger cellular immune responses after vaccination [66].

### 2.2. Types of Cationic Nanoparticles in Cancer Vaccines

Over the past decades, multiple types of cationic nanoparticles have been applied in cancer vaccines (Table 1). The composition of cationic nanoparticle-based cancer vaccines can be roughly divided into four categories; (i) type of antigen, (ii) cationic nanoparticle components, (iii) immune-stimulating adjuvants, (iv) additional excipients (Figure 1).

Lipid-based nanoparticles have been extensively used, which is no surprise since liposomes are one of the “oldest” and most clinically translated nanomaterials [90,91]. Additionally, efficient anti-tumor immunity has also been established with cationic polymer-based nanoparticles, hybrid nanoparticles and self-assembling nanoparticles composed of peptide conjugates (Table 1) [30,33,34,92]. Peptide conjugates are constructed of synthetic peptides (containing tumor epitopes) that are conjugated to charge-modifying molecules, such as ionizable polymers and specific amino acid sequences, resulting in controlled nanoparticle formation upon addition of an aqueous buffer to these conjugates [33,34,93]. Among the different vaccine formulations there are several examples where the cationic charge (or the cationic nanoparticle core) of the nanoparticle is shielded, e.g., by PEGylation or complexation with mRNA [9,94,95,96].

### 2.3. Types of Antigenic Molecules in Cationic Nanoparticulate Cancer Vaccine Formulations

Cancer vaccines aim to activate the cellular immune system, which plays a major role in anti-tumor immunity. DCs instruct and activate naïve T-cells by tumor antigen presentation and co-stimulation: the process called T-cell priming (Figure 2) [97,98,99]. In order to properly activate naïve T-cells, three different signals are required to be transmitted by the DCs: antigen presentation (signal 1), expressing of co-stimulatory molecules (signal 2) and production of co-stimulatory cytokines (signal 3) [22,99,100]. An effective cancer vaccine should be able to deliver tumor antigens to DCs and subsequently activate them, to ensure a proinflammatory immune condition to optimally induce functional antigen-specific T-cells [1,2,22,25]. Enhanced particle uptake and additional activation of DCs by a cationic delivery system could therefore be beneficial for the immunogenicity and efficacy of the vaccine. Most cancer vaccines will be composed of multiple epitopes, as tumors will have multiple mutations and thereby present a multitude of (mutated) antigens on their MHC molecules that are potentially recognizable for T-cells [19,20,101]. There are several options to include tumor-specific epitopes in cancer vaccines, such as antigens in the form of tumor lysates and full (mutated) proteins. However, both forms of antigens require a complex cGMP manufacturing process and are complex to formulate with cationic nanoparticles. In contrast, molecularly defined antigen types, such as nucleic acid sequences (mRNA/DNA) and short peptide sequences (<30 amino acids), can be relatively easy and fast-manufactured/synthesized in cell-free conditions under cGMP conditions. Therefore, these types of molecularly defined antigens are especially suitable for personalized cancer vaccines. Furthermore, these antigens have been formulated with a wide variety of cationic nanoparticles [9,29,36,61,62,63,64,66,67,87,102,103]. A great advantage of nucleotide-based vaccines is the low variability in physicochemical properties of the nucleic acids sequences when different epitopes are encoded [9]. Synthetic peptides have shown to be very effective and safe in cancer vaccines and have been used in multiple clinical trials [13,15,16,18,104,105]. Additionally, synthetic peptides offer the possibility for further chemical modifications to increase nanoparticulate peptide loading (e.g., lipopeptides), adjuvant conjugation and the development of self-assembling nanoparticles [33,34,75,93,106]. Additionally, there is extensive expertise with formulating peptides into nanoparticles [29,33,34,36,67]. In contrast, whole protein-based antigens and tumor lysates require production in/based on living cells resulting in more complex manufacturing and purification steps. This makes those antigens less suitable for multi-epitope vaccines. Despite this, whole proteins admixed with cationic liposomes have shown potent cellular immune responses upon vaccination in different studies [26,70,107]. Furthermore, several reports show tumor control when whole tumor extracts or tumor cell lysates have been formulated with cationic nanoparticles [77,78,108,109].

### 2.4. Antigen Classes in Cancer Vaccines

In cancer vaccines, three major antigen classes can be distinguished that are either derived from overexpressed self-antigens, tumor-associated pathogens, or based on DNA mutations, as summarized in Figure 3 and described below. The first cancer vaccines have been designed based on overexpressed self-antigens, but have had only limited clinical success [2,7,25,110]. Vaccines based on viral oncoproteins and tumor-specific DNA mutations have shown promising (pre-)clinical results and numerous efforts are ongoing in the clinical development of vaccines targeting such antigen classes [9,10,13,15,16,18,19,89,111].

#### 2.4.1. Tumor-Associated Antigens

Tumor-associated antigens (TAAs) can be derived from the expression of several gene classes: tissue-specific, tumor-testis, embryonal or genes that are upregulated in expression in cancer tissue as compared to healthy tissue [2,7,25,110,112]. The major limitation for vaccination with these antigen classes is the low immunogenicity of TAAs, since they are generally seen as self-antigens by the immune system. When breaking through tolerance and activated strongly, TAA-specific T-cells can also show unwanted “off-target” effects in healthy tissues expressing these genes, resulting in autoimmunity [25,110]. During the past decades, TAA-targeting cancer vaccines have had limited clinical success and several clinical trials did not continue after phase III [2,25].

#### 2.4.2. Viral Oncoproteins

Viral oncoproteins are only present in malignancies triggered by oncoviral infections, such as those caused by hepatitis B virus, human papillomavirus (HPV) and the Epstein-Barr virus (EBV). Prophylactic vaccines inducing virus-specific antibodies (humoral immune response) aim to prevent viral infection upon contact with the virus. This humoral response is, however, not effective against virus-infected cells or virally transformed cells in a (pre-)malignant disease state [15,113]. Established infection results in the incorporation of viral DNA or RNA in the host cell, resulting in the presentation of viral epitopes on the surface of the infected/malignant cell via MHC molecules, offering vaccine targets, which are tumor cell-exclusive. Multiple vaccines aiming to induce a T-cell mediated immune response directed towards viral oncoproteins are currently under preclinical and clinical evaluation [14,15,16,18,29,114,115,116].

#### 2.4.3. Neoantigens

Neoantigens result from somatic DNA mutations in tumors cells, resulting in a tumor exclusive set of antigenic peptide sequences (Figure 2). This class of cancer antigens is by definition non-self and will therefore have a strong potency to be immunogenic. Since these mutations are randomly induced, each patient will have a unique neoantigen profile, allowing development of personalized therapeutic cancer vaccines [2,12,20,21,25]. The unique expression of neoantigens in cancer cells and not in healthy cells will make such vaccines highly tumor-specific with little or no immune-related side effects expected. Upon neoantigen identification, the vaccine manufacturing time should be as short as possible, since these vaccines are for diagnosed cancer patients who decease in limited time without treatment [10,12,20]. Personalized cancer vaccines should be composed of multiple epitopes to induce a diverse set of antigen-specific T-cells. Multiple antigenic epitopes can be manufactured relatively fast under cGMP conditions in synthetic antigen formats (synthetic peptides, or antigen-encoding mRNA or DNA), which can be formulated relatively fast with cationic nanoparticles. Recent advancements in next-generation sequencing, bioinformatics and vaccine manufacturing have allowed for a rapid translation of neoantigen vaccines from murine models to the first clinical trials [9,10,13,19,33,34,89,105,111,117,118]. Delivery systems for these vaccines should be able to accommodate a wide variety of physicochemically distinct antigens (either as synthetic peptides, mRNA or DNA), since every patient has a unique set of antigens. Recent studies have shown that lipid-based nanoparticles meet these requirements and could therefore offer clinically applicable vaccine formulation platforms [9,10,36].

## 3. Biodistribution of Cationic Nanoparticulate Vaccines

The route of administration of nanoparticulate vaccines has shown to influence the quality and magnitude of cellular immune responses [9,26,29,34,119]. Many studies have been performed using different administration routes for cationic nanoparticulate vaccine formulations. However, the effect of the administration route can only be compared for the same nanoparticles, since the effects of particle composition, biodistribution and particle-specific effects on T-cell priming are not fully understood yet. Nonetheless, based on the current literature, several general biodistribution profiles and mechanisms of action have been related to specific administration routes, as summarized in Table 2.

After administration, cationic nanoparticles interact with a variety of (macro)molecules (e.g., proteins, lipids) that are present in the biological fluid at the site of injection (SOI). This interaction results in the coating of the nanoparticles, i.e., the formation of a so-called protein corona, resulting in a change of the particles’ physicochemical characteristics, which can result in particle deposition at the SOI [94,123]. Composition of the corona is influenced by factors, such as nanoparticle properties (e.g., size, charge and composition), administration route and composition of the biological fluid at the SOI [130,131,132]. Depot formation by cationic nanoparticles is reported for the intradermal (i.d.), subcutaneous (s.c.) and intramuscular (i.m.) routes and has been shown to result in prolonged antigen presence at the SOI and a sustained nanoparticle draining to the lymph nodes [34,67,120,121,122,123,133]. Both mechanisms have been related to prolonged antigen presentation by DCs and their subsequent activation, resulting in efficient tumor immunity in multiple preclinical mouse models [29,34,67,68]. Fluorescently labeled nanoparticles admixed with fluorescently labeled protein showed an increased retention at the SOI upon s.c. and i.m. injection compared to free protein. This depot, containing both the protein antigen and cationic nanoparticles, showed an increased immune cell infiltration compared to the free antigen [120,123]. When cationic particles were PEGylated, the depot formation decreased, most likely because of a decrease in electrostatic interactions between the nanoparticles and macromolecules in the biological fluid [94,95,96,120,134]. Self-assembling cationic nanoparticles, based on peptide conjugated to a charge-modifying amino acid sequence and a TLR 7/8 ligand, have been used to compare the induction of antigen-specific CD8^+^ T-cells upon i.v. and s.c. administration in mice [33,34]. The i.v. route resulted in a short burst exposure of antigen in the circulation, whereby particles were not detectable anymore after 24 h, while the s.c.-administrated nanoparticles could be detected up to 2 weeks at the SOI. The s.c. route resulted in the highest frequencies of antigen-specific CD8^+^ T-cells, while the i.v. route induced antigen-specific CD8^+^ T-cells that were less prone to exhaustion [9,33,34]. This indicates that depot formation for synthetic peptide-loaded nanoparticle-based vaccines results in higher levels of antigen-specific T-cells, compared to lesser depot formation. In our lab, cationic liposomes loaded with antigenic synthetic peptide resulted in superior anti-tumor immunity via i.d. administration in comparison to s.c. [29,66,67,68]. Therefore, this indicates that the i.d. route is most optimal for peptide-based immunization. This is most likely because the skin contains many DCs and Langerhans cells, which are key in processing and presenting antigens to T-cells. In our group, mechanistic studies are currently on-going to determine the biodistribution and depot formation of both the peptide and liposomes upon i.d. injection to obtain insight into the in vivo behavior of cationic liposomes after i.d. administration.

Upon i.v. administration of cationic nanoparticulate mRNA vaccines, functional tumor-specific CD8^+^ T-cells in both mice and man were efficiently induced [9,10,33,34,125]. The i.v. biodistribution was systematically studied with lipoplexes, composed of the cationic lipid DOTMA and the fusogenic lipid DOPE, containing mRNA encoding the firefly luciferase gene. The lipid:mRNA ratio was varied to produce lipoplexes with a net cationic, neutral and anionic charge, while all lipid formulations had the same lipid composition. The cationic nanoparticles accumulated in the lungs, while the anionic nanoparticles were mainly detected the spleen, as quantified by luciferase expression [9]. Neutral and near neutral lipoplexes were unstable and therefore not applied in vivo. The anionic nanoparticles were most likely filtered from the bloodstream by APCs in the spleen [9]. Besides, antigen-encoding mRNA was complexed with cationic liposomes in such a ratio that net anionic lipoplexes were yielded. These anionic lipoplexes efficiently induced antigen-specific T-cells that were able to regress tumors, as shown in multiple mouse models [9,14]. Interestingly, s.c. injection of mRNA-loaded lipoplexes induced lower levels of antigen-specific T-cells compared to the i.v. injection. These results indicate that a short burst exposure via i.v. administration is the most efficient administration route for T-cell induction for mRNA-based nanoparticulate cancer vaccines. Potentially the s.c. administered mRNA vaccines are less efficient in transfection due to depot formation and are thereby more prone to degradation. This is in apparent contrast to the required long exposure with synthetic peptide-loaded nanoparticles via the i.d. route. It is likely that mRNA vaccines will sustain expression of the antigenic polypeptides for an extended period of time upon i.v. administration.

These studies with mRNA lipoplexes demonstrate that also cationic nanoparticles from which the charge is fully shielded are very effective in the induction of antigen-specific T-cell responses [9,10,94,96,120].

The exact in vivo interactions between protein-coated cationic liposomes and immune cells largely remain a black box. However, upon injection, the protein corona will alter the physicochemical properties of the nanoparticles, potentially affecting their biodistribution [95,96]. The formation of protein–nanoparticle complexes upon i.d. and s.c. injection could be beneficial for peptide- and protein-based vaccines by promoting depot formation [95,96,107,135,136]. On the other hand, a potential risk is formed with i.v. injection of cationic nanoparticles, because larger complexes may be formed in the bloodstream, which in turn may lead to the blockade of capillaries, potentially resulting in thrombotic events [137,138]. A strategy to circumvent aggregation upon i.v. injection is by shielding the cationic charge of the nanoparticles by mRNA complexation or PEGylation [9]. Based on the literature, it is clear that cationic nanoparticles can establish anti-tumor immunity upon i.d., s.c. and i.m. administration, via depot formation, or exposure upon i.v. administration, via systemic antigen exposure. There is a limited number of studies that compared multiple administration routes for the same cationic nanoparticle-based cancer vaccine [9,29,33,34]. Such studies can help in understanding the role of the administration route in inducing tumor-specific T-cells and thereby expanding the clinical application of cancer vaccines.

## 4. Cationic Nanoparticles: Molecular Mechanism of Action

Cationic lipids and polymers incorporated in nanoparticles are known to have immunostimulatory properties and, if dosed appropriately, can be utilized to elicit strong antigen-specific immune responses [58,139,140,141,142]. These nanoparticles often consist of cationic lipids or polymers and neutral helper lipids/polymers. Specific cell cascade pathways in DCs have been described for different cationic particles upon uptake. Furthermore, it has been reported that cationic particles can influence cross-presentation, the process in which DCs present exogenous derived antigens in MHC-I molecules to CD8^+^ T-cells [31,44,143,144,145]. These two processes are described in detail below and are summarized in Figure 4. The protein corona could play a role in the nanoparticle’s behavior; however, the in vivo effects on the immunogenicity of cationic nanoparticles are not (yet) fully understood [130,131,132,137,138,146,147,148,149]. Therefore, we review known intracellular pathways below to gain insight in the mechanisms of action.

### 4.1. Immunostimulatory Effects of Cationic Lipids and Polymers in Nanoparticles

The immune-stimulating capacity of cationic nanoparticles is mediated via several molecular pathways and can enhance vaccine efficacy by activating DCs, resulting in a strong antigen-specific T-cell response. Multiple studies have shown upregulation of co-stimulatory molecules and, in some cases, increased production of proinflammatory cytokines after exposure of cationic nanoparticles to DCs [28,31,44,71,74,121,143,144,150,151]. Bone marrow-derived dendritic cells (BMDCs) incubated with DOTAP-containing cationic liposomes upregulated transcription of proinflammatory chemokine genes, the maturation marker CD11c and increased expression of co-stimulatory molecules CD80/CD86 both in vitro and in in vivo [28,71]. No or little activation was observed when the BMDCs were incubated with neutral or anionic liposomes [28]. Authors report that upregulation is mainly regulated by the extracellular signal-regulated kinase (ERK) pathway which is activated by reactive oxygen species (ROS). The DOTAP-based liposomes were shown to trigger low levels of ROS, which have been associated with DC maturation. However, when the DOTAP concentration exceeded 200 µM, high levels of apoptosis were observed due to abundant ROS production [71,74,152]. A different study showed that an increase in surface charge density of DOTAP:DOPC liposomes increased DC maturation in vitro, indicating that there is a fine balance between immune stimulation by cationic lipids and cell death by high DOTAP concentrations [57]. These studies did not report about differences in protein coronas of the studied liposomes. The differences in charge or surface charge density could influence nanoparticle uptake and could thereby affect the immunogenicity. For several cationic lipids, the molecular structure has been related to DC activation. BMDCs exposed to cationic liposomes containing lipids with a quaternary amine headgroup upregulated CCL2 transcription, while BMDCS incubated with DODAP, a DOTAP analog with a tertiary amine headgroup, did not. For DOTAP, the in vivo immunological activity has even shown to be higher for the (R)-enantiomer than the (S)-enantiomer [73]. These results suggest that there is a receptor-specific interaction of DOTAP within DCs that is involved in immune cell activation. Liposomes containing the cationic lipid diC14-amidine were able to induce transcription and secretion of a wider range of proinflammatory cytokines (IL12p40, TNF-α) as well as upregulation of CD80/CD86 via TLR-4 activation [44,73,150]. It has been reported that TLR-4 activation by diC14-amide is not mediated via the LPS, the natural TLR-4 ligand, binding sites. Authors reported TLR-4 activation by a diC14-amide-mediated dimerization [153]. Cationic lipopolyamines with saturated C18 tails have been shown to activate the human TLR-2 receptor [31,143,151]. Due to the structural diversity of cationic lipids and different immune effects, it is likely that multiple receptors and pathways are involved in their mechanism of action [144,145].

Compared to cationic lipids, less research has been done on the immune-stimulating capacity of cationic polymers. Direct TLR activation as well as membrane destabilization has been reported in the literature as potential mechanism. The in vivo activation of the TLR-4 receptor in mouse macrophages and splenocytes by cationic polymers, such as polyethylenimine (PEI), polylysine and cationic dextran, has been reported. Based on the structural differences between these polymers and LPS, it is not expected that TLR-4 activation occurs via the LPS binding site. Activation could be mediated via a similar mechanism as diC14-amide, but this has not (yet) been reported. Polycationic polymers have also been shown to form holes in cell membranes, resulting in immune cell activation. These holes could be formed by the hydrolysis of cell-membrane phospholipids, mediated by the cationic groups in the polymers. An increase in the degree of cationic groups was related to increasing immune stimulating capacity [27,43,154,155,156].

### 4.2. Enhanced Cross-Presentation by Cationic Lipids and Polymers in Cationic Nanoparticles

Several studies have reported that cationic nanoparticles composed of cationic lipids and/or polymers can enhance antigen delivery to the cytosol and improve cross-presentation by DCs [42,51,103,157,158,159]. BMDCs incubated with antigen-loaded cationic liposomes increased proliferation of antigen-specific CD8^+^ T-cells in vitro, while no differences in proliferation were observed after incubation with anionic liposomes. The cationic liposomes interfered with the acidification of lysosomes, resulting in less acid conditions that reportedly reduced antigen degradation and destabilized lysosomal membranes, resulting in increased cytosolic antigen delivery. In this study, cationic lipids containing a quaternary amine, DOTAP, or tertiary amine, DC-cholesterol, were used and Gao et al. hypothesized that the amine groups were responsible for a lesser decrease in lysosomal pH [157]. In an older study with cationic microparticles, it also has been reported that cationic particles interfere with the acidification of phagosomes in DCs [55]. The ROS production has recently been linked to destabilization of lysosomal membranes, which results in antigen leakage to the cytosol, resulting in an increased cross-presentation [160]. Induced ROS production by cationic liposomes in DCs could improve cross-presentation via this mechanism as well. Endo/lysosomal membrane destabilization by cationic polymers has been described by the protonation of functional groups, often amines, resulting in polymer swelling and membrane destabilization (also referred to as the proton sponge effect in the literature). Upon endocytosis, several polymers can bind protons in the endosomal fluid, which increases electrostatically repulsons in the polymer, resulting in the swelling. The buffering capacity of the polymers also prevents acidification of the endosome, resulting in an increased activity of the V-ATPase pump and chloride channels leading to an influx of ions. As a result, the osmotic pressure rises, which contributes further to membrane instability of the endosome. Because of the destabilized membrane, the antigen can leak into the cytosol, which can improve antigen cross-presentation [42,161,162]. Nanoparticles composed of PEI mixed with OVA protein, in varying ratios, were shown to improve antigen cross-presentation compared to the free protein in mouse BMDCs in vitro. Authors have reported that the cross-presentation is improved by the proton–sponge effect and is related to the cationic nature of the polymer [162]. Further identification of the functional groups involved in the proton–sponge effect can further help the design of nanoparticles that deliver antigens to the cytosol.

## 5. Conclusions and Perspectives

Cationic nanoparticulate cancer vaccine formulations are very promising platforms for specific immunotherapy of cancer. Such nanoparticulate formulations can be used in synthetically produced antigens (peptides, mRNA, DNA) as multi-epitope vaccines and readily produced under cGMP conditions. The wide variety of studied cationic nanoparticles has revealed mechanisms by which the cancer vaccine efficacy is improved: efficient antigen uptake, molecular activation of APCs and distinct biodistribution profiles. Vaccine administration in the skin is of special interest, since the skin contains relatively large amounts of DCs and is easily accessible for drug delivery. The in vivo efficacy of cationic nanoparticle-based cancer vaccines is determined by the interplay of particle characteristics, administration route and subsequent handling by the immune system. Systematic immunological studies with cationic, neutral and anionic nanoparticles can further increase our understanding of optimal vaccine delivery. Next to charge, the shape, size and rigidity of nanoparticles play a role in vaccine efficacy and offer possibilities to further improve the design of cationic nanoparticle-based cancer vaccines [163,164,165]. The extensive research efforts that are on-going in the tumor immunology field rapidly expand our mechanistic understanding of tumor-specific T-cell biology allowing further fine-tuning of therapeutic cancer vaccine design. Altogether, cationic nanoparticle-based cancer vaccines hold great potential for near-future cancer immunotherapy in patients.

## Figures and Tables

**Figure 1 pharmaceutics-13-00596-f001:**
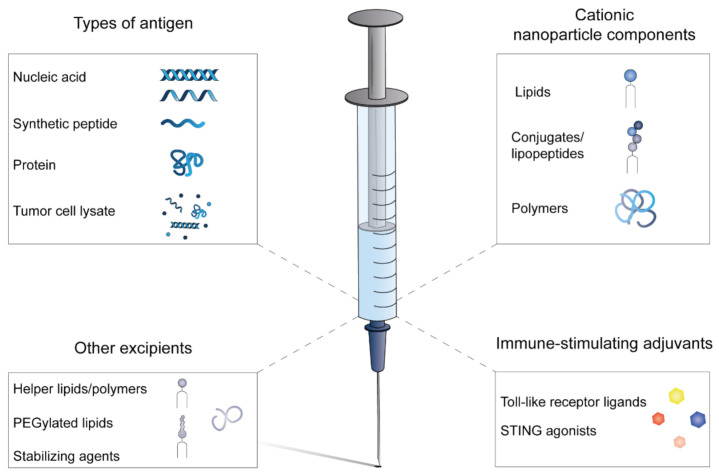
**Composition of cationic nanoparticle-based cancer vaccines.** Whole-tumor antigens have been incorporated as whole protein or tumor cell lysate in cancer vaccines. Nucleic acids encoding tumor antigens or synthetic peptides can be synthetically manufactured under cGMP conditions. The cationic component of the nanoparticles is often combined with neutral helper lipids and/or polymers to manufacture stable nanoparticles and optimize intracellular antigen delivery by incorporation of fusogenic molecules. In most formulations, additional immune-stimulating adjuvants are included to ensure sufficient APC activation. Stabilizing agents, such as sugars, buffers and surfactants, are included to formulate a stable vaccine that can be stored and transported.

**Figure 2 pharmaceutics-13-00596-f002:**
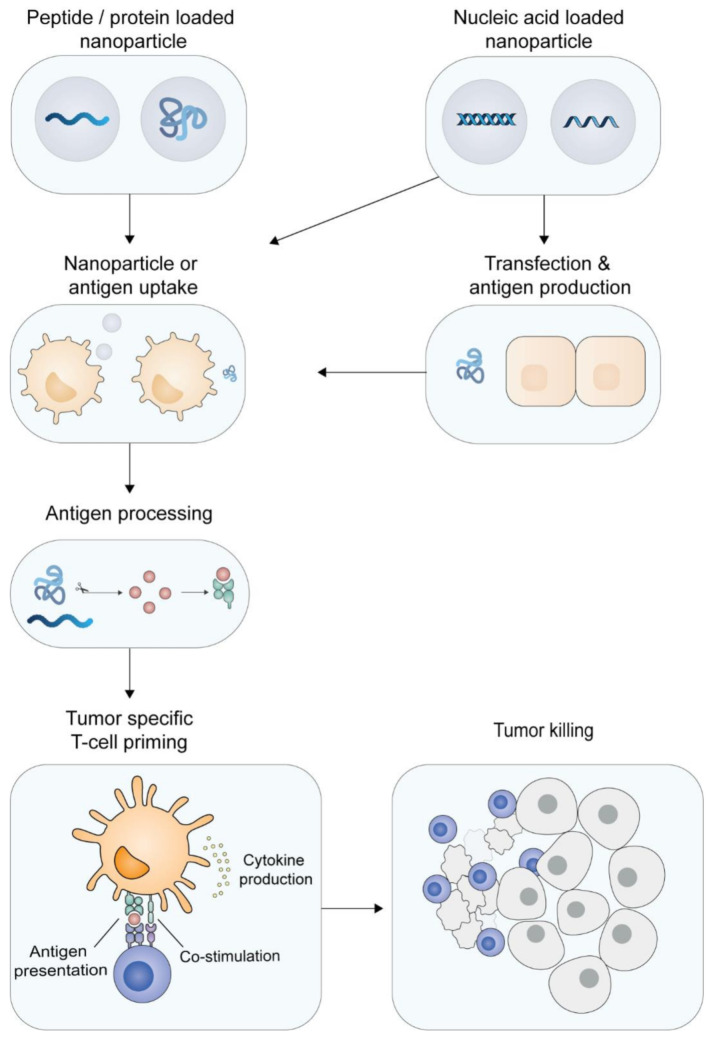
**Priming of tumor-specific T-cells.** Dendritic cells can engulf synthetic peptide- or protein-loaded cationic nanoparticles and subsequently process the particles. Nucleic acid-loaded particles can also transfect non-immune cells (like epidermal or muscle cells) that, upon transcription and translation, produce antigenic proteins, which are subsequently taken up by DCs. The antigen is processed and the tumor-specific epitopes are presented by the DC to CD8^+^ T-cells (cross-presentation) or to CD4^+^ T-cells. In combination with immune stimulation, the DCs upregulate co-stimulatory molecules and produce pro-inflammatory cytokines, resulting in priming of tumor-specific T-cells. The activated tumor-specific T-cells are able to home to the tumor tissue and recognize and kill the malignant cells.

**Figure 3 pharmaceutics-13-00596-f003:**
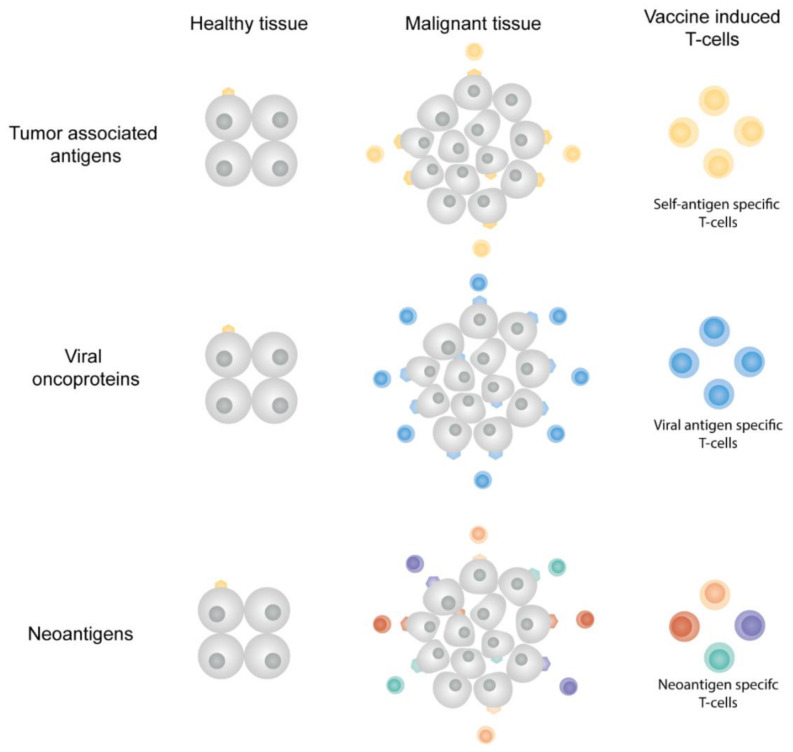
**Antigen classes in cancer vaccines.** Tumor-associated antigens are self-antigens that can be (over)expressed in tumor tissues. Vaccine-induced TAA-specific T-cells can kill both tumor cells and healthy cells. Viral oncoproteins are uniquely expressed by malignant cells in which the viral transformation resulted in tumor growth. Neoantigens originate from DNA mutations present in the cancerous cells, the neoantigens are therefore only expressed in malignant tissue. Tumor tissues often express multiple neoantigens, offering multiple vaccine targets and multiple neoantigen-specific T-cell populations.

**Figure 4 pharmaceutics-13-00596-f004:**
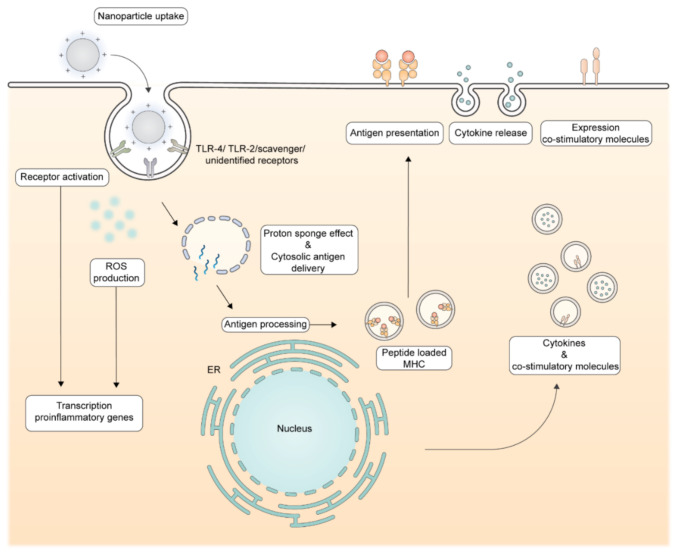
**Molecular immune-stimulating mechanisms by cationic nanoparticles.** Upon uptake, cationic nanoparticles can have immune-stimulating properties via the induction of reactive oxygen species (ROS) and receptor activation, such as the TLR-4. These pathways result in the transcription of pro-inflammatory genes, resulting in translation of proinflammatory cytokines and co-stimulatory molecules. The proton–sponge effect results in cytosolic antigen delivery, enabling the polypeptide antigen to enter the antigen processing machinery. Proteasome and peptidase mediated processing will deliver oligopeptides to be presented in MHC molecules which will be transported to the cell surface. An increase in cytosolic antigen delivery combined with the immune-stimulating properties of cationic nanoparticles results in efficient priming of antigen-specific T-cells.

**Table 1 pharmaceutics-13-00596-t001:** Currently reported cationic nanoparticle-based cancer vaccine formulations.

Study Type	Antigen Source	Particle Type	Cationic Component	Type of Antigen	Molecular Adjuvant	Administration Route	Reference
Murine							
Preclinical	Ovalbumin	Liposomes	DOTAP	Peptide	Poly [I:C]	i.d.	[67]
Preclinical	TC-1 & Melanoma (B16-F10)	Liposomes	DOTAP	Peptide	Poly [I:C]	i.d.	[29,68]
Preclinical	TC-1 & Melanoma (B16-OVA)	Liposomes	DDA	Protein Peptide	Poly [I:C]	i.p.	[69,70]
Preclinical	TC-1	Liposomes	DOTAP	Peptide Lipopeptide	n.a.	s.c.	[71,72,73,74,75]
Preclinical	Melanoma (B16-F10)	Liposomes	DOTAP	Peptide	n.a.	s.c.	[76]
Preclinical	Hepatoma (Hepa 1–6)	Liposomes	DOTAP	Tumor lysate	Poly [I:C]	i.p.	[77]
Preclinical	Glioma (GL261)	Liposomes	TMAG	Tumor extract	n.a.	i.p.	[78]
Preclinical	Melanoma (B16-BL6)	Liposomes	DOTMA	Plasmid DNA	Mannose	i.p.	[79]
Preclinical	Colon carcinoma (CT-26)	Liposomes & w/o/w emulsion	DC-Chol	Peptide	Pam2Cys	p.o.	[80]
Preclinical	Melanoma (B16-F10 & BPD6)	LCP nanoparticles	DOTAP	Peptide mRNA	Mannose CpG	s.c.	[81,82,83]
Preclinical	Breast cancer	LCP nanoparticles	DOTAP	mRNA	Mannose	s.c.	[84]
Preclinical	Colon carcinoma (CT-26)	LCP nanoparticles	DOTAP	mRNA	Mannose CpG cGAMP	s.c.	[85]
Preclinical	Thymic lymphoma (E.G7-OVA)	Lipid-polymer nanoparticles	DOTAP	Protein	Mannose Imiquimod MPLA	s.c.	[86]
Preclinical	Thymic Lymphoma (E.G7-OVA)	Lipid-polymer Nanoparticles	Non disclosed lipid	mRNA	n.a.	i.v.	[87]
Preclinical	Ovalbumin	Polymer-based	Methacrylated dextran	Peptide	Poly [I:C]	i.d.	[30]
Preclinical	TC-1, Melanoma, Colon carcinoma (B16-OVA & CT-26)	Lipoplexes	DOTMA	mRNA	n.a.	i.v.	[9,14]
Preclinical	Colon carcinoma (MC-38 & CT-26)	Lipoplexes	DOTMA	Peptide	CpG	i.v. & s.c.	[35]
Preclinical	Colon carcinoma (MC-38), Melanoma (B16-F10 & B16.OVA), TC-1	Self-assembling nanoparticles	Amino acid sequence	Peptide	Imidazoquinoline- based TLR 7/8a	i.v. & s.c.	[33,34]
Human							
Phase I	Melanoma	Lipoplexes	DOTMA	mRNA	n.a.	i.v.	[9]
Phase I	Prostate cancer	Liposomes	DDA	Peptide	Poly [I:C]	i.p. & i.m.	[88]
Phase I/IIa	Melanoma, NSCLC, Bladder cancer	Liposomes	DDA	Peptide	Poly [I:C]	i.p.& i.m.	[89]

Abbreviations: LCP = lipid-Calcium-Phosphate, CpG = synthetic oligodeoxynucleotides containing CpG motifs, MPLA = monophosphoryl lipid A, TLR = toll-like receptor ligand, NSCLC = non-small-cell small lung cancer, i.d. = intradermal, i.v. = intravenous, i.p = intraperitoneal, s.c. = subcutaneous, i.m. = intramuscular, p.o. = oral.

**Table 2 pharmaceutics-13-00596-t002:** Routes of administration for cationic nanoparticulate cancer vaccines with the accompanied observed biodistribution. A limited number of studies describe i.n. administration of cationic nanoparticles, but not with cancer vaccines. Nonetheless, the i.n. route with cationic nanoparticulate formulations with peptide and mRNA have been included in the table.

Route of Administration	Biodistribution Profiles	Ref
Intradermal (i.d.)	- Depot formation at the SOI - Prolonged Ag presentation at the SOI - Prolonged Ag presentation in draining lymph nodes	[29,62,67,68]
Subcutaneous (s.c.)	- Depot formation at the SOI - Prolonged Ag presentation at the SOI - Prolonged Ag presentation in draining lymph nodes	[62,71,73,75,94,95,96,107,120]
Intramuscular (i.m.)	- Depot formation at the SOI - Prolonged Ag presentation at the SOI	[95,96,107,121,122,123]
Intraperitoneal (i.p.)	- Rapid drainage to multiple lymphoid organs - Limited/no depot formation	[26,62,69,89,124]
Intravenously (i.v.)	- Systemic Ag exposure - Uptake by splenic DCs - Uptake by APCs in lungs	[9,10,33,34,62,125,126]
Intranodal (i.n.)	- High vaccine concentration in lymph nodes - Complex injection (ultra-sound or tracer guided)	[119,127,128,129]

Abbreviations: Ag = antigen, SOI = site of injection.

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
