# Peer review of "Cationic Nanoparticle-Based Cancer Vaccines"

_pharmaceutics, 2021, doi:10.3390/pharmaceutics13050596_

Round 1

Reviewer 1 Report

The review is well-written and well-organized. the topic is very important especially in cancer immunotherapy era. the authors summerized the available data hihhlighting also the unmet needs. 

only three minor points:

  1. line 91: it is not clear the eaning of "T-cell vaccines". please, explaine it.
  2. line 183: maybe "oncolytic pathogens" is not appropited, did the authors mean "tumor-associated pathogens"? If yes, maybe also EBV needs to be mentioned.
  3. lines 242-243: the meaning of the sentence is not clear

Author Response

Reviewer: The review is well-written and well-organized. the topic is very important especially in cancer immunotherapy era. the authors summarized the available data highlighting also the unmet needs. 

Authors: We thank the reviewer for the positive words about our manuscript. 

only three minor points:

  1. line 91: it is not clear the meaning of "T-cell vaccines". please, explain it.

Authors: With T-cell vaccines we mean cancer vaccines that aim to induce T-cell mediated tumor immunity. We have changed ‘’T-cell vaccines’’ by ‘’cancer vaccines’’ for clarity.

  1. line 183: maybe "oncolytic pathogens" is not appropited, did the authors mean "tumor-associated pathogens"? If yes, maybe also EBV needs to be mentioned.

Authors: We thank the reviewer for the suggestion and substituted oncolytic pathogens by tumor-associated pathogens. In line 202 we have added EBV, the authors agree with the reviewer that this virus can result in a malignancies and is therefore relevant for the manuscript. 

  1. lines 242-243: the meaning of the sentence is not clear

Authors: We thank the reviewer for pointing this out. We noted that the introductory sentences of section 3 were not present in the submitted version of the manuscript, making sentence 242-243 unclear. We have added these sentences again, by wich sentence 242 – 243 should be clear now. 

Reviewer 2 Report

Regarding the van der Maaden et al. pharmaceutics-1186869, it indeed represents an informative and intriguing review on cationic nanoparticle-based cancer vaccines. It is very suitable for pharmaceutics. The following suggestion may help improve the manuscript. But none of my further comments will disqualify it for publication:

  1. The authors should generally introduce the barriers for in vivo vaccines delivery, such as insufficient uptake and endosomal escape of negatively charged nucleic acid and protein and nuclease degradation of nucleic acid, to clarify the necessity of using nanomedicine (here is cationic nanoparticle).
  2. The general mechanism of cationic nanoparticle formation should be introduced. I think in this review, most cases are electrostatic complexation.
  3. I totally agree with the point in this nice review that specific cargo and target will determine the results. Especially, "Interestingly, s.c. injection of mRNA-loaded lipoplexes induced lower levels of antigen specific T-cells compared to the i.v. injection. These results indicate that a short burst exposure via i.v. administration is the most efficient administration route for T-cell induction for mRNA-based nanoparticulate cancer vaccines. This is in apparent contrast to the required long exposure with synthetic peptide-loaded nanoparticles via the i.d. route." In fact, we also observed s.c. injection of polyplex mRNA micelles is less effective. Could you kindly give some comments on this point? It may be due to fast degradation of mRNA when stucked in injection site (positive charge).
  4. In fact, inducing immunogenic cell death in situ should also be a effective strategy for vaccines. The authors should give brief discussion on this and cover some research (Int. J. Mol. Sci. 2018, 19, 594; Angewandte Chemie International Edition, 2020, 59(32): 13526-13530).
  5. In conclusion, the authors give the perspectives mainly from biology. The authors should also give some perspectives from materials science, such as how to further improve in vivo functionality of cationic nanoparticle as vaccines using stimuli-responsive strategies and actively-targeted strategies (e.g., J. Am. Chem. Soc. 2021, 143, 538−559).

Author Response

Regarding the van der Maaden et al. pharmaceutics-1186869, it indeed represents an informative and intriguing review on cationic nanoparticle-based cancer vaccines. It is very suitable for pharmaceutics. The following suggestion may help improve the manuscript. But none of my further comments will disqualify it for publication:

Authors: We thank the reviewer for the kind words and suggestions .

  1. The authors should generally introduce the barriers for in vivo vaccines delivery, such as insufficient uptake and endosomal escape of negatively charged nucleic acid and protein and nuclease degradation of nucleic acid, to clarify the necessity of using nanomedicine (here is cationic nanoparticle).

Authors: We have added an extra sentence on the in vivo barriers in lines 94 – 96.

  1. The general mechanism of cationic nanoparticle formation should be introduced. I think in this review, most cases are electrostatic complexation.

Authors: Cationic nanoparticle preparation and particle formation is outside the scope of this manuscript, as there is already a lot of literature/reviews availavle on this subject. Therefore we have chosen not to include cationic nanoparticle preparation and particle formation.

  1. I totally agree with the point in this nice review that specific cargo and target will determine the results. Especially, "Interestingly, s.c. injection of mRNA-loaded lipoplexes induced lower levels of antigen specific T-cells compared to the i.v. injection. These results indicate that a short burst exposure via i.v. administration is the most efficient administration route for T-cell induction for mRNA-based nanoparticulate cancer vaccines. This is in apparent contrast to the required long exposure with synthetic peptide-loaded nanoparticles via the i.d. route." In fact, we also observed s.c. injection of polyplex mRNA micelles is less effective. Could you kindly give some comments on this point? It may be due to fast degradation of mRNA when stucked in injection site (positive charge).

Authors: It is very interesting to read that experiments by the reviewer support our conclusions in the manuscript. We added some additional discussion to our point in lines 307-312.

  1. In fact, inducing immunogenic cell death in situ should also be a effective strategy for vaccines. The authors should give brief discussion on this and cover some research (Int. J. Mol. Sci. 2018, 19, 594; Angewandte Chemie International Edition, 2020, 59(32): 13526-13530).

Authors: Indeed cell death can improve immune activation of DCs by introducing a scala of chemoattractants etc., which can improve the magnitude of immune responses against the co-administered antigen. In this manuscript we have specifically focussed on the molecular processes and pathways of activation in DCs upon the uptake of cationic nanoparticles, which does not apply to apoptotic cells and was therefore not discussed. 

  1. In conclusion, the authors give the perspectives mainly from biology. The authors should also give some perspectives from materials science, such as how to further improve in vivo functionality of cationic nanoparticle as vaccines using stimuli-responsive strategies and actively-targeted strategies (e.g., J. Am. Soc. 2021, 143, 538−559).

Authors: The manuscript is focused on the application of cationic nanoparticle-based cancer vaccines and their biological mechanisms of action. Therefore, we have decided not to include the perspectives from material sciences to maintain a clear focus in the manuscript.